# Inference of Gene Regulatory Network Uncovers the Linkage between Circadian Clock and Crassulacean Acid Metabolism in *Kalanch**oë fedtschenkoi*

**DOI:** 10.3390/cells10092217

**Published:** 2021-08-27

**Authors:** Robert C. Moseley, Francis Motta, Gerald A. Tuskan, Steven B. Haase, Xiaohan Yang

**Affiliations:** 1Department of Biology, Duke University, Durham, NC 27708, USA; robert.moseley@duke.edu (R.C.M.); steve.haase@duke.edu (S.B.H.); 2Department of Mathematical Sciences, Florida Atlantic University, Boca Raton, FL 33431, USA; fmotta@fau.edu; 3Biosciences Division, Oak Ridge National Laboratory, Oak Ridge, TN 37831, USA; tuskanga@ornl.gov; 4The Center for Bioenergy Innovation, Oak Ridge National Laboratory, Oak Ridge, TN 37831, USA; 5Department of Medicine, Duke University, Durham, NC 27708, USA

**Keywords:** circadian clock, drought, crassulacean acid metabolism, Local Edge Machine, stomata, gene expression

## Abstract

The circadian clock drives time-specific gene expression, enabling biological processes to be temporally controlled. Plants that conduct crassulacean acid metabolism (CAM) photosynthesis represent an interesting case of circadian regulation of gene expression as stomatal movement is temporally inverted relative to stomatal movement in C3 plants. The mechanisms behind how the circadian clock enabled physiological differences at the molecular level is not well understood. Recently, the rescheduling of gene expression was reported as a mechanism to explain how CAM evolved from C3. Therefore, we investigated whether core circadian clock genes in CAM plants were re-phased during evolution, or whether networks of phase-specific genes were simply re-wired to different core clock genes. We identified candidate core clock genes based on gene expression features and then applied the Local Edge Machine (LEM) algorithm to infer regulatory relationships between this new set of core candidates and known core clock genes in *Kalanch**oë fedtschenkoi*. We further inferred stomata-related gene targets for known and candidate core clock genes and constructed a gene regulatory network for core clock and stomata-related genes. Our results provide new insight into the mechanism of circadian control of CAM-related genes in *K. fedtschenkoi*, facilitating the engineering of CAM machinery into non-CAM plants for sustainable crop production in water-limited environments.

## 1. Introduction

The circadian clock is a vital time-keeping mechanism that synchronizes periodic environmental signals to an organism’s physiology, allowing for biological processes to function in a timely manner. This mechanism is very important in plants due to their sessile nature. Numerous environmental signals and stressors to plants are cyclic, such as light availability, temperature, and predation. The circadian clock thus enables plants to activate the appropriate processes in response to these repeating variables.

Plants that photosynthesize through crassulacean acid metabolism (CAM) are great examples of how plants synchronize biological processes to their environment. CAM plants exhibit improved photosynthetic efficiency due to a temporal separation of CO_2_ fixation and improved water-use efficiency due to the inverted day/night pattern of stomatal opening and closing, relative to C3 plants. In short, CAM plants open their stomata during the night allowing for uptake of atmospheric CO_2_ and close their stomata during the night when normal photosynthetic processes occur [1]. This inversion of stomatal movement (i.e., the opening and closing of stomata) is also an important drought avoidance/tolerance mechanism in CAM plants, by which water loss caused by evapotranspiration is decreased. These traits coupled with the global issue of increased frequency and intensity of drought [2,3] have generated an increase in CAM research with the goal of engineering these traits into C3 plants, to enable better drought responses and/or improved drought tolerance [4,5]. Currently, it is theorized that the temporal separation of CO_2_ fixation is under control of the circadian clock [6,7] and that the inverted stomatal movement could be a result of a change in clock regulation [8]. However, the events that lead to these drastic physiological differences seen in CAM plants via the circadian clock are not well understand.

To explain potential ways the circadian clock was involved in the evolution of CAM, the first step is to understand components that make up the circadian clock network. The circadian clock is a complex network that has at its center a small regulatory network of core clock genes, generally referred to as the core clock [9]. Core clock genes are defined as highly connected transcription factors (TFs) which subsequently create positive and negative feedback loops. This network of interlocking feedback loops causes the core clock genes to be rhythmically expressed. Connected to the core clock are additional genes, usually TFs, as the core clock TFs regulate not only themselves but also genes outside the core clock. This transmits the rhythmicity of gene expression generated by the core clock to additional networks, resulting in these specific networks to have rhythmically expressed genes. These genes and the networks they are in are generally referred to as clock-regulated (Appendix A). Eventually, the phenotypes connected to these networks display rhythmicity as well. Using these definitions one can being to generate testable hypotheses on how the circadian clock could have been involved in the evolution of CAM.

For example, stomatal movement has been shown to be under the control of the circadian clock [10], therefore, the inversion of stomatal movement seen in CAM plants, relative to C3, could have occurred from rewiring between the core clock and the gene regulatory network (GRN) controlling stomatal movement. Specifically, the stomatal movement GRN could be under the control of another core clock gene in the core clock (Appendix A), or the original core clock gene was rewired within the core clock network (Appendix A), altering timing in the stomatal movement GRN.

A more intriguing hypothesis is that CAM plants use different genes than C3 plants in the core clock network. This isn’t to say that the CAM core clock network is constructed differently or consist of functionally different genes, but rather has conserved network topology and functionally similar, non-orthologous genes. This hypothesis is based on the idea that network topology is as equally, if not more important in GRNs as the network components themselves [11,12]. The conservation of topology and sequence divergence in components in circadian clock network across species has been well documented [9,13,14]. This presents the idea that the stomatal movement GRN in CAM plants could be regulated by unknown core clock genes.

To test these hypotheses, construction of gene network models that incorporate the underlying temporal dynamics is needed. Traditional methods to build models, such as ChiP-chip, ChiP-seq, and mutant expression profiling, can be laborious and can miss the dynamics of the network. Fortunately, high-throughput technologies have allowed for tractable methods of measuring transcription levels in time-course experiments [15,16,17,18,19,20,21]. These data exhibit the underlying temporal dynamics of gene expression and new computational tools have taken advantage of this property to help infer and build functional gene network models [22,23]. 

Therefore, we utilized time-course transcriptome data from the CAM plant *Kalanchoë fedtschenkoi* [21] to infer the regulatory relationships between the core clock network and stomatal-related genes. Through network inference, several genes were identified as potentially new core clock genes in *K. fedtschenkoi*. Additionally, stomata-related genes, including genes with rescheduled gene expression, were predicted to be regulated by core clock genes in *K. fedtschenkoi*. The circadian clock plays a crucial role in the physiological response to various environmental stresses in plants, such as drought [24] and our results provide a circadian clock network model to experimentally test various hypotheses on circadian control of stomatal movement in CAM. Fully elucidating the links between the circadian clock and CAM will be key for successful engineering of CAM into C3 plants for improved drought response and tolerance.

## 2. Materials and Methods

### 2.1. Time-Course Gene Expression Data

The diel gene expression data for *K. fedtschenkoi* and *Arabidopsis thaliana* were obtained from [21] and [25], respectively. The *K. fedtschenkoi* expression data were collected at 2, 4, 6, 8, 10, 12, 14, 16, 18, 20, 22, and 24 h whereas the *A. thaliana* data were collected at 4, 8, 12, 16, 20, and 24 h after the starting of the light period. Since the *A. thaliana* gene expression data was measured at 4-h intervals and the *K. fedtschenkoi* data was measured at 2-h intervals, the *A. thaliana* data was adjusted to arrive at expression profiles for all *A. thaliana* and *K. fedtschenkoi* genes on the same time scale. Here, the piecewise cubic Hermite interpolating polynomial (pchip) interpolation function in the pandas Python library was used to sample the *A. thaliana* data to simulate gene expression levels at additional time points so that both time-course data sets consisted of the same time intervals: 2, 4, 6, 8, 10, 12, 14, 16, 18, 20, 22, and 24 h after the starting of the light period. Pchip was preferred over the more common method of cubic spline interpolation due to cubic spline’s tendency to overshoot which introduces oscillations. Additionally, pchip maintains the shape of the data and has been used on microarray time course data sets [26,27]. Additionally, *K. fedtschenkoi* genes with a max FPKM<1 were considered noise and removed. The rhythmic *K. fedtschenkoi* and *A. thaliana* gene sets were taken from [28].

### 2.2. Identifying Candidate Core Clock Genes

Identifying core clock genes using time-course data can be difficult due to similarities in their gene expression profiles with the gene expression profiles of clock-regulated genes. However, core clock genes in various species have been found to have the highest amplitudes and the most statistically significant rhythms [22,29,30]. The periodicity detection methods of de Lichtenberg (DL) [31,32] and JTK-CYCLE (JTK) [33,34] are periodicity detection algorithms that take into account the amplitude of time-course gene expression and if the period of expression matches to the period length in question, respectively. Therefore, to better identify core clock genes, a new metric has been established, termed DLxJTK, that combines these two features of DL and JTK [35]. DLxJTK uses the *p*-values for amplitude from DL and for periodicity from JTK for each gene and has been used in mammalian, fungal and plant systems with success. The DLxJTK formula is
(1)$$DLxJTK=P_{per}\;P_{amp}\left({\left(1+\frac{P_{per}}{0.001}\right)}^{2}\right)\left({\left(1+\frac{P_{amp}}{0.001}\right)}^{2}\right),$$
where *P_per_* is the JTK *p*-value for periodicity and *P_amp_* is the DL *p*-value for amplitude. The output of DLxJTK is a ranked ordered list of genes, with core clock genes being near the top of the list. DLxJTK was applied to the *K. fedtschenkoi* rhythmic gene list from [28] and the top 60 TFs were used for further analysis. 

Previously, a method, Local Edge Machine (LEM), was described that enabled the discovery of new components of the mouse circadian clock network [22,36]. To identify high-confidence core clock genes in *K. fedtschenkoi*, LEM was used to infer regulatory relationships between the top 60 candidate core clock TFs and TFs orthologous to known core clock genes in *A. thaliana* (from here on referred to as “known core clock TFs”) in two steps. Firstly, LEM was used to identify if any of the 60 candidate core clock TFs were regulated by the known core clock TFs. The known core clock transcriptional activators used were reveille 8 (RVE8), RVE6, light-regulated WD 1 (LWD1), LWD2, night light-inducible and clock-regulated 1 (LNK1), and LNK2 [37]. The known core clock transcriptional repressors used were circadian clock associated 1 (CCA1), late elongated hypocotyl (LHY), timing of cab expression 1 (TOC1), CCA1 hiking expedition (CHE), LUX, NOX, pseudo-response regulator 9 (PRR9), PRR7, PRR5, ELF3, and ELF4 [37]. LUX is only active after forming the evening complex with early flowering 3 (ELF3) and 4 (ELF4) [38,39], which do not bind to DNA [40]. ELF3 and ELF4 were included for this reason. LEM was set to only use the respective mode of gene regulation for each known core clock TF used. All candidate core clock TFs were set as targets for the known core clock TFs. 

Secondly, since core clock TFs are known to regulate other known core clock genes, LEM was run again but with the top 60 candidate core clock TFs as potential regulators of the known core clock TFs. Both modes of regulation (activation and repression) were allowed for each candidate TF in LEM. Allowing both modes produces a probability of regulation for each mode between a candidate TF and a known core clock TF. The mode of regulation with the highest probability of the two was used for subsequent analyses and visualizations. To identify high-confidence candidate core clock TFs, a measure of likelihood, described in [22], was used for each candidate core clock TF. This measure is calculated by taking the maximum LEM probability that the candidate core clock TF was a regulator of any known core clock TF and multiplying it by the maximum LEM probability that the candidate core clock TF was regulated by any known core clock TF.

### 2.3. Identify Core Clock-Regulators of Stomata-Related Genes

LEM was applied to identify potential regulatory relationships between core clock TFs and stomata-related genes. Known core clock TFs plus the candidate core clock TFs were used as potential regulators of stomata-related genes. Stomata-related genes in *K. fedtschenkoi* were identified as genes that are orthologous to an *A. thaliana* gene that is either annotated as stomata-related or is known as stomata-related. Orthology between species was based on placement within the same ortholog group. Additionally, new *K. fedtschenkoi* stomata-related genes identified in [8] were included as well.

### 2.4. Gene Ontology Analysis

Gene Ontology (GO) terms for the *K. fedtschenkoi* and *A. thaliana* were obtained from Phytozome v12.1 [41]. *K. fedtschenkoi* genes encoding putative transcription factors were retrieved from [21]. Using ClueGO [42], observed GO biological process were subjected to the right-sided hypergeometric enrichment test at medium network specificity selection and *p*-value correction was performed using the Holm-Bonferroni step-down method [43]. There was a minimum of 3 and a maximum of 8 selected GO tree levels, while each cluster was set to include a minimum of between 3% and 4% of genes associated with each term. GO term fusion and grouping settings were selected to minimize GO term redundancy and the term enriched at the highest level of significance was used as the representative term for each functional cluster. The GO terms with *p*-values less than or equal to 0.05 were considered significantly enriched.

### 2.5. Comparative Analysis of Gene Expression

To calculate time-delay between time-course gene expression profiles, the diel expression data were normalized by Z-score transformation. Pair-wise circular cross correlation was calculated for the orthologous gene pairs of interest for all possible time delays using the SciPy library [44] in Python. Circular cross correlation produces a correlation coefficient between two genes (e.g., gene 1 and gene 2) as a function of the lag. With each correlation coefficient, a lag value was given. The lag values were then converted into hours, giving an estimate on time delay. The time delay at which the correlation was maximum was selected as the estimated delay between the two genes. Spearman’s rank correlation coefficient was then calculated between gene 1’s expression data and the shifted expression data of gene 2 by its estimated time delay.

## 3. Results

### 3.1. Candidate Core Clock Transcription Factors in Kalanchoë fedtschenkoi

DLxJTK was used to rank the rhythmic *K. fedtschenkoi* gene list from [28] to pull out potential core clock genes. The top 60 TFs ranked by DLxJTK were selected from the full list of DLxJTK ranked genes (Appendix A) and were used as candidate core clock TFs. The candidate core clock TFs covered a majority of the phases of the day and displayed a bimodal distribution with peaks occurring before subjective night and before subjective morning (Appendix A). These results are consistent with phase call distributions of circadian genes seen in other plant species, as well as non-plant species [15,16,45]. To determine if any of the *K. fedtschenkoi* TFs were orthologous to *A. thaliana* TFs that have been annotated as circadian-related, ortholog groups (OGs) constructed in [21] were investigated. Only 36 of the 60 *K. fedtschenkoi* TFs were placed in OGs with 75 *A. thaliana* genes. Of the 75 *A. thaliana* genes, 13 were found to be associated with circadian rhythm (Appendix A). Several TFs belonged to the C2H2, MYB-HB, and C2C2-CO families containing 15, 9, and 9 genes, respectively (Appendix A).

LEM was employed next to infer if any of the known core clock TFs could regulate the 60 candidate core clock TFs. After applying a cutoff of 0.7 to remove low probability regulatory relationships, all known core clock TFs were predicted to regulate at least one candidate core clock TF (Figure 1). One ortholog of LNK2 (Kaladp0099s0129) was predicted to regulate 20 candidate clock-regulated TFs while the orthologs of LUX (Kaladp0033s0047) and LNK1 (Kaladp0607s0046) were predicted to regulate eight different candidate core clock TFs each. All but 6 candidate core clock TFs were found to be activated or repressed by known core clock TFs, while 13 candidate core clock TFs were found to activate or repress known core clock TFs. Only one candidate core clock TF (Kaladp0748s0043) was predicted to regulate more than one known core clock TF (Figure 2). LEM was employed again to infer if any of the 60 candidate core clock TFs could regulate the known core clock TFs. The LEM probabilities of a known core clock TF regulating a candidate core clock TF were used with the LEM probabilities of a candidate core clock TF regulating a known core clock TF to compute likelihood measures (see Material and Methods). Using a likelihood ranking cutoff of 0.7, eight candidate core clock TFs were identified high-confidence core clock TFs in *K. fedtschenkoi* (Figure 1A and Appendix A). The eight high-confidence candidate core clock TFs were phased to three separate phases of the day (i.e., morning, midday, evening). The phasing of the candidate core clock TFs and their regulatory relationships in the context of the core circadian clock model can be seen in Figure 1B. To annotate each of the eight high-confidence candidate core clock TFs, *A. thaliana* orthologs were identified by placement in OGs. All *A. thaliana* orthologs identified were rhythmic [28]. Descriptions of the genes are below and separated into three categories corresponding to phase of the day (i.e., morning, midday, evening) of max gene expression as follows.

### 3.2. Morning Phased Candidate Core Clock Transcription Factors

Three of the high-confidence candidate core clock TFs (Kaladp0011s1342, Kaladp0009s0042, and Kaladp1154s0002) in *K. fedtschenkoi* were phased to the morning (Table 1). Among these three, Kaladp0011s1342 and Kaladp0009s0042 were not placed in an OG with an *A. thaliana* gene. Therefore, their respective protein sequence was used to search the NCBI non-redundant protein BLAST database using an E-value cutoff of 1e-5. Kaladp0011s1342 was phased to 2 h before the beginning of the light period and found to have a similar protein sequence with two *A. thaliana* proteins, AT3G58120 (BZIP61) and AT2G42380 (BZIP34) (Table 1 and Appendix A). BZIP TFs are known to regulate pathogen defense, light and stress signaling, seed maturation and flower development [46]. BZIP34 has been predicted to be involved in the regulation of lipid metabolism and/or cellular transport [47]. BZIP34 and BZIP61 were both rhythmic and were phased to four and eight h after the beginning of the light period, respectively (Table 1 and Appendix A). Kaladp0009s0042’s protein sequence lacked homology with any protein sequences in *A. thaliana*. The protein sequence of Kaladp0009s0042 was found to contain a Dof (DNA-binding with one finger) domain. Additionally, the remaining *K. fedtschenkoi* gene in this group, Kaladp1154s0002, was found in an OG containing three *A. thaliana* genes encoding for the Dof domain-containing proteins, including cycling DOF factor 1 (AT5G62430; CDF1), 2 (AT5G39660; CDF2), and 3 (AT3G47500; CDF3). CDF1, CDF2, and CDF3 are involved in various signaling pathways, including photoperiodic and light signaling, stress responses and circadian clock regulation [48]. CDF1 transcription has been reported to be repressed by the circadian clock pseudo-response regulator protein family [49,50,51,52,53] and activated by the circadian clock genes CCA1 and LHY [54], resulting in CDF1 gene expression at dawn. All three *A. thaliana* orthologs were rhythmic and phased to dawn (Table 1 and Appendix A). CDF1 protein accumulation is also regulated by the circadian clock through protein stability via complex formation with gigantean (GI) or flavin-binding, Kelch repeat, F-Box 1 (FKF1) [48,55,56]. However, feedback into the clock has not been reported for the CDFs. Both Kaladp0009s0042 and Kaladp1154s0002 gene expression peaked at dawn (Table 1 and Appendix A). 

### 3.3. Midday Phased Candidate Core Clock Transcription Factors

Four of the high-confidence candidate core clock TFs (Kaladp0878s0025, Kaladp0674s0030, Kaladp0748s0043, and Kaladp0674s0182) in *K. fedtschenkoi* were phased to midday (Table 1 and Appendix A). Kaladp0878s0025 had one *A. thaliana* ortholog (AT1G07050), which encodes for a Constans-like protein encoding gene and is a predicted target of the clock regulator GI [57]. The transcript expression of Kaladp0878s0025 was phased to 8 h after the beginning of the light period, whereas its *A. thaliana* ortholog had gene expression phased to 12 h after light (Table 1 and Appendix A).

Kaladp0674s0030 had two *A. thaliana* orthologs, AT5G63160 and AT3G48360, with both encoding for members of the Bric-a-Brac/Tramtrack/Broad (BTB) gene family, specifically BT1 and BT2, respectively. Only BT2 had gene expression data in the [25] dataset and was found to be rhythmic with gene expression phased to 20 h after light. Kaladp0674s0030 was phased to six h after light, displaying a shift in expression between the two species (Table 1 and Appendix A). BT2 is known to activate telomerase expression in mature *A. thaliana* leaves, play a critical role in nitrogen-use efficiency in *A. thaliana* and *Oryza sativa*, suppress sugar and ABA responses, and positively regulate certain auxin responses in plants [58,59]. Additionally, BT2 is regulated diurnally and controlled by the circadian clock, with maximum expression in the dark [58]. It has been suggested that the pattern of gene expression for BT2 mRNA could be linked to the availability of photosynthate, which is a product of photosynthesis [59].

Kaladp0748s0043 was found to be orthologous to the *A. thaliana* plant homeobox family protein BELL1 (BEL1), which is a key regulator of ovule development and needed for auxin and cytokinin signaling pathways for correct patterning of the ovule [60]. Kaladp0748s0043’s gene expression was phased to 6 h after light, whereas BEL1 in *A. thaliana* was phased to 12 h after light (Table 1 and Appendix A). Lastly, Kaladp0674s0182 had only one *A. thaliana* ortholog, AT3G29270, which encodes for a ring/U-box superfamily protein. Proteins in this superfamily are involved in protein ubiquitination. 

Kaladp0674s0182 was orthologous to RING/U-box superfamily proteins in *A. thaliana*, which are typically E3 ubiquitin ligases. Due to this ambiguity in function, Kaladp0674s0182 will not be used in further analysis.

### 3.4. Evening Phased Candidate Core Clock Transcription Factors

Only one high-confidence candidate core clock TF (Kaladp0007s0017) in *K. fedtschenkoi* had gene expression phased to the evening (Table 1 and Appendix A). Kaladp0007s0017 had two *A. thaliana* orthologs, which encode for jasmonate (JA)-associated MYC2-like proteins 1 (AT2G46510; JAM1) and 2 (AT1G01260; JAM2). JAM1 has been reported as the balancing component opposite of the MYC2 TF in the JA signaling pathway [61]. Specifically, JAM1 and MYC2 are induced by JA and share many of the same target genes. Where MYC2 activates transcription of multiple genes, including JAM1, JAM1 negatively influences gene expression by physically interfering with MYC2 binding to promoter regions of target genes [61]. The target genes for both TFs are considered "early-responsive JA genes" as changes in gene expression of target genes occur within 1 h of JA detection [62]. JA signaling is linked to activation of defense pathways and subsequent stomatal closure and has been reported to be under the control of the circadian clock through regulation of MYC2 via repression of transcription [63]. Additionally, JAM1 has been reported to participate in ABA signaling as a positive regulator as overexpression of the gene in *A. thaliana* increased drought tolerance [63]. JAM1 and JAM2 in *A. thaliana* had gene expression phased to 12 and 9 h after light, respectively, whereas Kaladp0007s0017 had gene expression phased to 14 h after light (Table 1 and Appendix A).

### 3.5. Core Clock Regulation of Stomata-Related Genes in Kalanchoë fedtschenkoi

The fact that stomatal movement has been inverted in CAM plants raises the question whether the circadian clock, specifically core clock TFs, played a role in this inversion through rescheduling of gene expression. To investigate this question, regulatory relationships were inferred between the seven high-confidence candidates, plus the known core clock TFs in *K. fedtschenkoi* (Appendix A), and the 1,605 stomata-related gene in *K. fedtschenkoi*, which were identified as rhythmic in a separate study [8]. Two high-confidence candidate core clock TFs in *K. fedtschenkoi*, Kalado0033s0047 and Kaladp0032s0115, were found on the target list (Appendix A of [8]) of rhythmic stomata-related genes and were subsequently removed as targets. Additionally, high-confidence candidate clock TFs Kaladp0007s0017, Kaladp1154s0002, Kaladp0674s0030, Kaladp0878s0025, and Kaladp0007s0017 were also in the target list and removed as targets.

LEM was used to infer regulatory relationships between core clock TFs and rhythmic stomata-related genes. Using a cutoff of 0.7 on the LEM output related to the probability of a TF regulating a gene, 582 of the 1,605 stomata-related genes were inferred to be regulated by core clock TFs (Appendix A and Appendix A). A visualization of the overall network can be seen in Figure 3. Core clock genes are known to activate or repress genes, depending on their mode of regulation, during specific phases of the day. For instance, CCA1 and LHY repress genes that are expressed during the evening [37]. To determine if LEM predicts the appropriate phase of regulation for core clock genes, the phase calls of target genes for core clock TFs with ≥10 target genes were examined in diel plots (Figure 4 and Figure 5). LNK1 and LNK2 are thought to activate gene expression of targets during the afternoon and evening [37], and in line with this, a majority of the predicted stomata-related gene targets for both LNK genes in *K. fedtschenkoi* were phased to the afternoon and evening (Figure 4). PRR7 is known to repress genes during dawn and in the morning [37]. Indeed, predicted target genes of PRR7 in *K. fedtschenkoi* were phased to dawn and the morning (Figure 4). Lastly, ELF4 is known to repress genes in the morning and evening by forming a complex with ELF3 and LUX [38]. LEM only predicted morning-phased gene targets for ELF4 in *K. fedtschenkoi* (Figure 4). The remaining components of the evening complex were examined as well to see if any of their targets were phased to the evening. None of the remaining evening complex components, including the two other ELF4 genes, had target genes phased to the evening (Appendix A). Additionally, ELF4, ELF3, and LUX did not share similar targets.

Candidate core clock TFs are allowed in the LEM model to be activators or repressors. Most TFs were inferred to be both activators or repressors, so it is unclear whether they are acting as one or the other (Appendix A). However, high-confidence candidate clock TFs Kaladp0007s0017 and Kaladp9878s0025 were primarily predicted as activators of gene expression and candidate core clock TF Kaladp1154s0002 was predicted primarily as a repressor of gene expression (Appendix A). A majority of the targets for all high-confidence candidate core clock TFs were phased to either dusk or dawn (Figure 5).

To determine what biological functions in stomata-related processes are under the control of the circadian clock, enrichment of associated gene ontology terms was performed. A majority of the biological functions enriched in the 582 rhythmic stomata-related genes were associated to protein phosphorylation (Figure 6 and Appendix A). To determine if any *K. fedtschenkoi* genes are related to *A. thaliana* genes annotated or known as stomata-related, OGs were examined again. Within the 582 *K. fedtschenkoi* genes, 49 were placed in OGs that contained *A. thaliana* genes that were either annotated or known as stomata-related genes (Appendix A). All the candidate core clock TFs and four known core clock TFs were predicted to regulate at least one of the 49 *K. fedtschenkoi* genes. The remaining stomata-related genes were identified in a separate study [8] as new stomata-related genes and all known and candidate core clock TFs were predicted to regulate at least one new stomata-related gene (Appendix A).

### 3.6. Regulation of Rescheduled Stomata-Related Genes

Five stomata-related genes identified in [21] were inferred to be regulated by core clock TFs, with only one of them being orthologous to a known stomata-related gene in *A. thaliana* (Table 2). Twelve stomata-related genes identified in [8] were inferred to be clock-regulated, with only three being orthologous to annotated or known stomata-related genes in *A. thaliana* (Table 2). Several of these genes were predicted to encode for protein kinases and transporters. High-confidence candidate core clock TF Kaladp0011s1342 was predicted to regulate the most stomata-related genes that displayed re-scheduling (Table 2). Interestingly, Kaladp0011s1342 also displayed rescheduling of gene expression relative to its two orthologs in *A. thaliana* (Appendix A) and xone of the targets of Kaladp0011s1342 was the rescheduled catalase 2 gene identified in [8].

## 4. Discussion

Through a combination of a new metric to identify potential core clock genes and the gene regulation inference algorithm, LEM, this study predicted several novel candidate core clock TFs in *K. fedtschenkoi*. These seven new core clock candidates predicted in *K. fedtschenkoi* are located in the transcriptional feedback loops (Figure 1B), which are consistent with the feedback-loop architecture of the core clock [37]. *A. thaliana* orthologs of several *K. fedtschenkoi* candidate core clock TFs have been reported to be connected to the circadian clock and involved in various signaling pathways. Additionally, most of the candidate core clock TFs were either phased to the morning or evening (Table 1), consistent with other reported circadian genes [15,16]. The discovery of these new potential core clock candidates supports our previous hypothesis that there could be unknown circadian genes in *K. fedtschenkoi* [21]. Recently, LEM was used to identify new core circadian clock genes in mouse [22], in which four out of the top ten genes were validated as clock-regulated genes via RNAi knockdown approach. The success of this application of LEM in mouse makes the candidate core clock TFs identified here in *K. fedtschenkoi* high-confidence candidates for future experimental work.

Appendix A illustrates two models explaining how the circadian clock could alter gene expression and therefore physiology of an organism. The first model (Appendix A) illustrates that a different core clock gene could have taken over the regulation of a physiological process, thus changing when the process occurs. The second model (Appendix A) illustrates that a change in the timing of a physiological process could be a result of the core clock gene, that regulates the process, being rewired in the core clock network. Evidence for both models explaining how stomatal movement was inverted in CAM plants by the circadian clock was found, suggesting that both mechanisms could have aided in the evolution of CAM. For instance, ELF4 was found to have rescheduled expression relative to its *A. thaliana* ortholog [28] and in the current study, ELF4 was predicted to regulate a rescheduled stomata-related gene (Table 2), in line with the second model (Appendix A).

It has been suggested that there could be new core clock genes in *K. fedtschenkoi* [21] and in this study, several TFs were predicted with high confidence to be new core clock genes in *K. fedtschenkoi* (Table 1 and Figure 3). This presented a new model by which the circadian clock could have altered stomatal movement in CAM plants via the new core clock genes regulating stomatal-related genes. Several rescheduled stomatal-related genes were inferred to be regulated by the predicted core clock genes (Table 2), supporting the model of new core clock genes regulating stomatal movement. For example, a duplicated CAT2 gene in *K. fedtschenkoi* was found to have rescheduled gene expression and proposed to be involved in the inversion of stomatal movement in *K. fedtschenkoi* [8]. The rescheduled CAT2 gene was inferred to be regulated by the predicted core clock gene Kaladp0011s1342. Evidence supporting this model and the two models illustrated in Appendix A suggest that the core clock played a role in reversing the day/night stomatal movement pattern in CAM photosynthesis species in comparison with C3 photosynthesis species through a combination of the three mechanisms.

Alternatively, the candidate core clock TFs may be used by the core clock network to integrate rhythmicity into the various signaling pathways they control. Experimental work is needed to determine their essentiality in the clock. For instance, protoplast transient reporter gene expression assay [64] can be used to validate the role of these TFs in the regulation of circadian rhythm. Specially, promoter fusion constructs can be used that contain the promoter of the candidate core clock TF driving the transcription of a fluorescent protein. Transfection of protoplasts and subsequent recording of fluorescence over 48 hours would enable a quick means to determine involvement with the clock. A *K. fedtschenkoi* protoplast protocol has not been published, though an *A. thaliana* protoplast assay may work based on the concept of core clock genes being highly connected. An additional study could examine the impact of the candidate core clock TF on the core clock network. Here, a two-promoter construct would be made with one promoter being a constitutive promoter driving the transcription of the respective candidate TF and the second promoter being the promoter of a known core clock gene that the candidate TF is predicted to target. The known core clock promoter would then drive the transcription of a fluorescent protein. If the candidate core clock TF is integrated into the core clock network, expression of the fluorescent protein will be altered relative to the expression of the core clock gene associated with the core clock promoter driving the fluorescent protein. 

Within the clock-regulated stomata-related genes, several GO terms associated with phosphorylation were significantly enriched (Figure 6). Additionally, several rescheduled stomata-related genes were identified as protein kinases (Table 2). Phosphorylation allows for rapid regulation of protein function and is known to play a significant role in stomatal movement. Furthermore, an extensive array of phosphorylation and dephosphorylation events occur in guard cells [65]. Signaling pathways for stomatal closure, e.g., ABA, and for stomatal opening, e.g., blue light, both involve protein kinases phosphorylating anion channels (ABA signaling: [66,67,68,69]) and H+-ATPases (blue light signaling: [70,71,72]). Evidence of gene expression rewiring of protein kinases involved in stomatal movement has also been identified in the CAM plant *Agave americana* [18]. Taken together, these results suggest that the clock played a role in the inversion of stomatal movement, potentially by rescheduling phosphorylation events of stomata-related genes. The direct substrates of these kinases and how they affect stomatal movement remains unknown. However, the channels and ATPase identified in Table 2 serve as good candidates to test as substrates for the protein kinases identified here.

Through gene regulatory network analysis, this study predicted a set of novel TFs that could be important components of either the core clock or the networks attached to the core clock in the CAM species *K. fedtschenkoi*. These candidate core clock TFs, if validated by experiments in the future, would significantly advance our understanding of the regulatory mechanism in CAM systems. Also, our analysis of the regulatory relationship between core clock TFs and stomata-related genes revealed that clock-facilitated rescheduling of protein kinases involved in stomatal movement aided the inverted stomatal movement seen in CAM plants, via connecting with different core clock TFs. These results provide new knowledge to inform genetic improvement of drought resistance in C3 photosynthesis plants for sustainable food and bioenergy production on dry and marginal lands.

## Figures and Tables

**Figure 1 cells-10-02217-f001:**
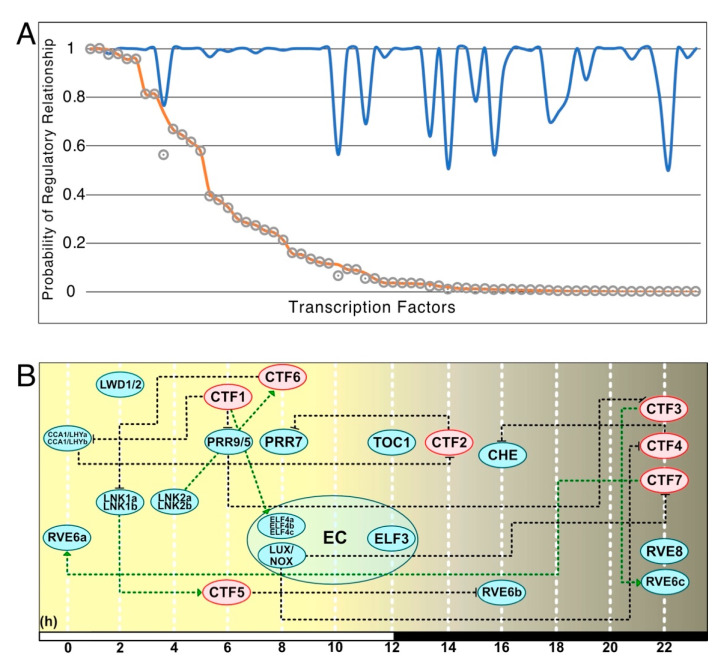
Seven candidate core clock transcription factors (CTF1-7) are predicted to play a role in the core clock network of *Kalanchoë fedtschenkoi*. (**A**) Predicted regulatory relationships between candidate core clock transcription factors and known core clock genes via the Local Edge Machine [22]. Blue line is the probability that any known core clock transcription factor regulates a candidate clock transcription factor. Orange line is the probability that a candidate core clock transcription factor regulates any known core clock transcription factor. Grey circle is the probability of a regulatory relationship. (**B**) The seven candidate core clock transcription factors and their relationship with the core clock network in *K. fedtschenkoi*. Blue ovals represent core clock genes and red ovals represent candidate clock transcription factors. The numbers arrayed at the bottom of the image indicate the number of hours passed after first light. Green dotted lines represent activation of transcription and black dotted lines represent repression of transcription. White and black bars indicate daytime (12-hour) and nighttime (12-hour), respectively. CTF: Candidate core clock transcription factor, EC: Evening complex.

**Figure 2 cells-10-02217-f002:**
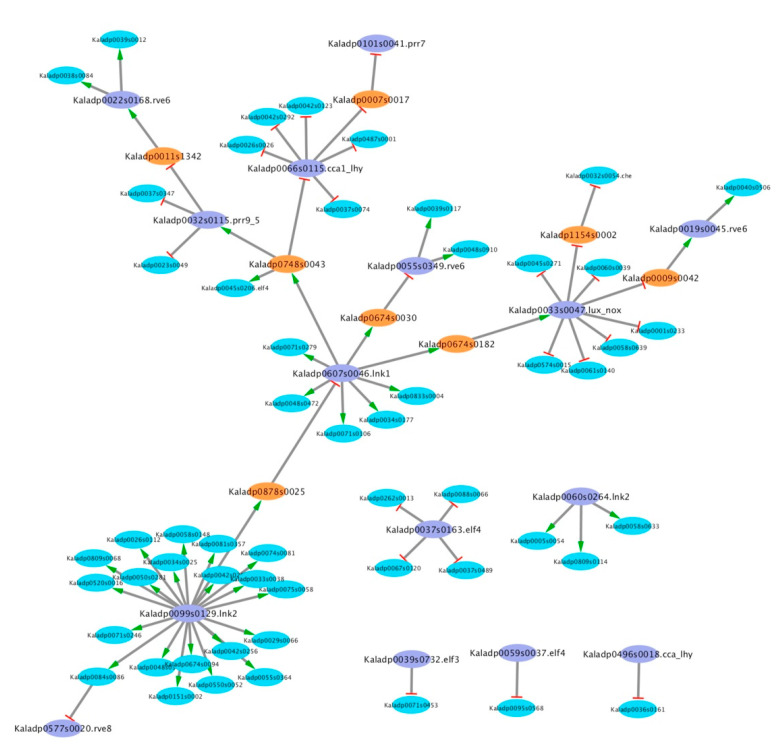
Several candidate core clock transcription factors are predicted by LEM to regulate and be regulated by known core clock transcription factors. Determination of mode of regulation is described in Materials and Methods. Orange ovals represent candidate core clock transcription factors predicted to regulate and be regulated by known core clock transcription factors. Purple ovals represent known clock transcription factors. Blue ovals represent candidate core clock transcription factors that are predicted to only be regulated by known core clock transcription factors. Edges with a green arrow represent activation of gene expression. Edges with red lines represent repression of gene expression.

**Figure 3 cells-10-02217-f003:**
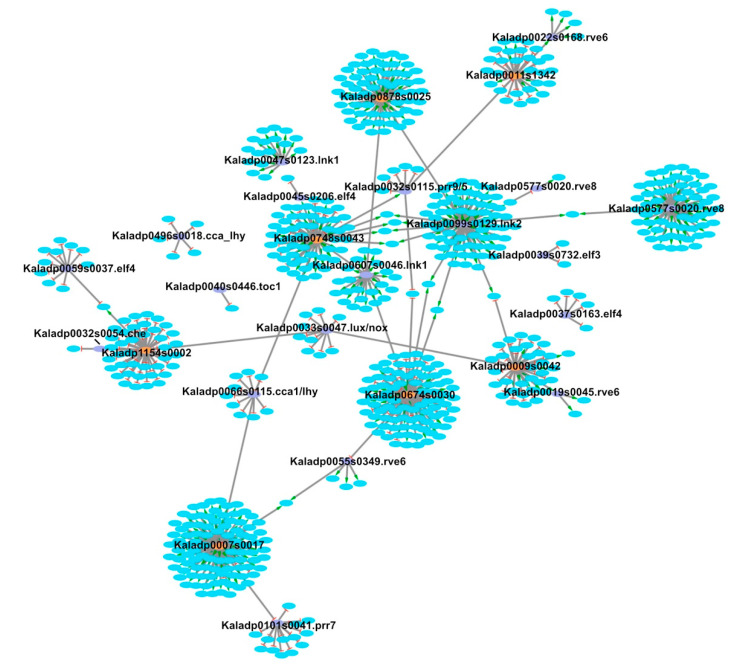
*Kalanchoë fedtschenkoi* core clock transcription factors and stomata-related genes regulatory network. Orange ovals represent candidate core clock transcription factors predicted to regulate and be regulated by known core clock transcription factors. Purple ovals represent known clock transcription factors. Blue ovals represent stomata-related genes. Edges with a green arrow represent activation of gene expression. Edges with red lines represent repression of gene expression.

**Figure 4 cells-10-02217-f004:**
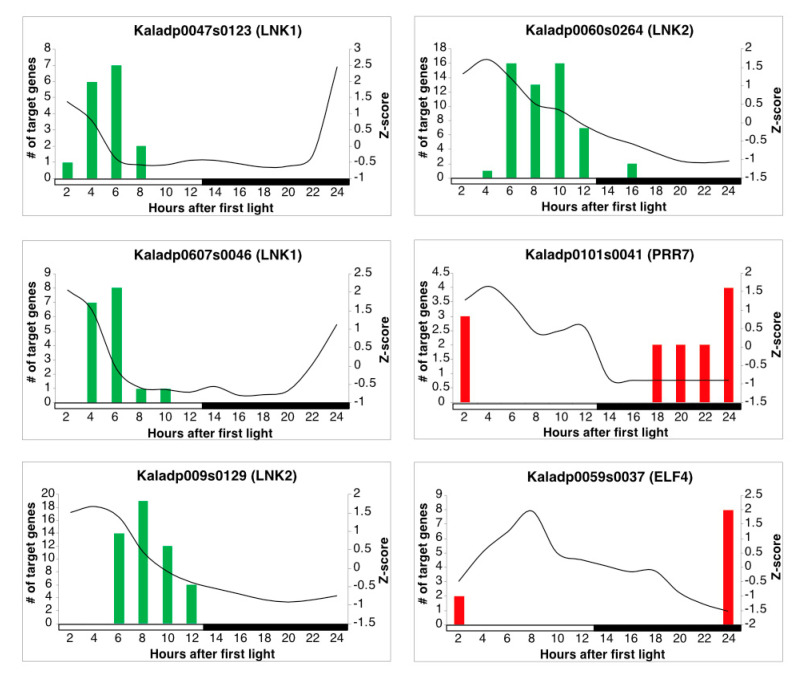
Gene expression profiles of *Kalanchoë fedtschenkoi* known core clock transcription factors and phase calls of their respective targets. The black lines represent the z-score standardized expression profile of the respective gene. Red and green bars represent the number of predicted target genes phases to the same time of day. Red bars signify that the target genes are repressed by the respective regulator and green bars signify that the target genes are activated by the respective regulator. White and black bars indicate daytime (12-hour) and nighttime (12-hour), respectively.

**Figure 5 cells-10-02217-f005:**
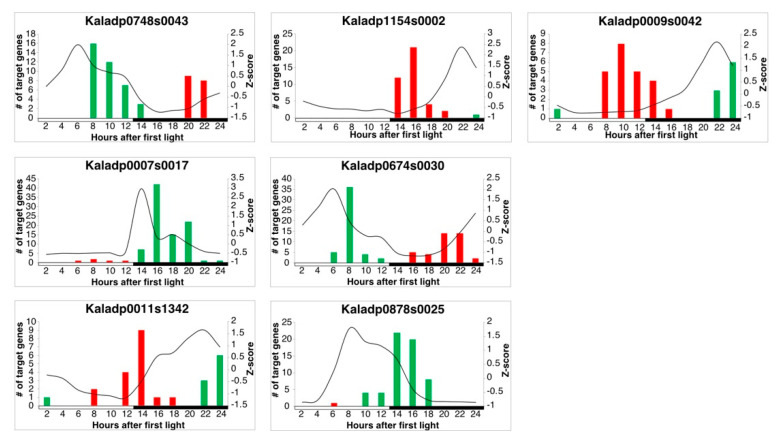
Gene expression profiles of *Kalanchoë fedtschenkoi* candidate core clock transcription factors and phase calls of their respective targets. The black lines represent the z-score standardized expression profile of the respective gene. Red and green bars represent the number of predicted target genes phases to the same time of day. Red bars signify that the target genes are repressed by the respective regulator and green bars signify that the target genes are activated by the respective regulator. White and black bars indicate daytime (12-hour) and nighttime (12-hour), respectively.

**Figure 6 cells-10-02217-f006:**
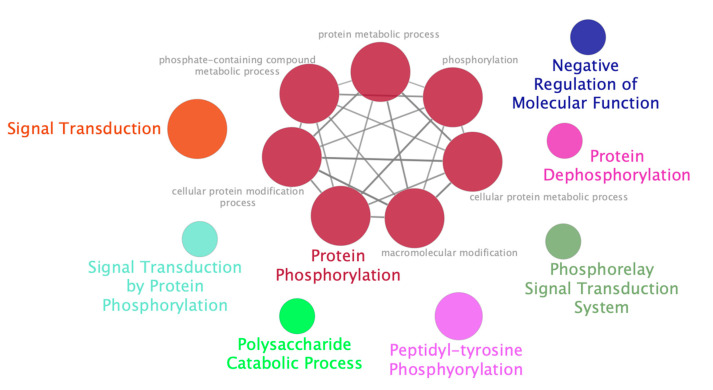
Enriched functional groups within of clock-regulated stomata-related genes in *Kalanchoë fedtschenkoi*. Enriched functional groups were determined in the Cytoscape application ClueGO. Parameters used in ClueGO are described in the Materials and Methods section. Colors represent individual functional groups. The sizes of circles are based on each GO term’s Bonferroni-corrected p-value, where a smaller *p*-value will have a larger circle. The associated *p*-values can be found in Supplemental Appendix A.

**Table 1 cells-10-02217-t001:** Candidate core clock transcription factors in *Kalanchoë fedtschenkoi*.

Kafe Gene ID	Kafe Phase Call	Arth Ortholog ID	Arth Gene Symbol	Arth Gene Desc.	Arth Phase Call	Kafe Shift ^1^ (hrs)	Spear Shift ^2^
Kaladp0748s0043 (CTF1)	6	AT5G41410	BEL1	POX (plant homeobox) family protein	12	−6	0.97
Kaladp0007s0017 (CTF2)	14	AT2G46510	JAM1	ABA-inducible BHLH-type transcription factor	12	+2	0.91
Kaladp0007s0017 (CTF2)	14	AT1G01260	JAM2	basic helix-loop-helix (bHLH) DNA-binding superfamily protein	8	+6	0.83
Kaladp0011s1342 (CTF3)	22	AT2G42380	BZIP34	Basic-leucine zipper (bZIP) transcription factor family protein	4	−8	0.96
Kaladp0011s1342 (CTF3)	22	AT3G58120	BZIP61	Basic-leucine zipper (bZIP) transcription factor family protein	8	−8	0.97
Kaladp1154s0002 (CTF4)	22	AT3G47500	CDF3	cycling DOF factor 3	2	−4	0.97
Kaladp1154s0002 (CTF4)	22	AT5G62430	CDF1	cycling DOF factor 1	24	−2	0.97
Kaladp1154s0002 (CTF4)	22	AT5G39660	CDF2	cycling DOF factor 2	24	−2	0.87
Kaladp0674s0030 (CTF5)	6	AT5G63160	BT1	BTB and TAZ domain protein 1			
Kaladp0674s0030 (CTF5)	6	AT3G48360	BT2	BTB and TAZ domain protein 2	20	+10	0.94
Kaladp0878s0025 (CTF6)	8	AT1G07050		CCT motif family protein	12	−2	0.92
Kaladp0009s0042 (CTF7)	22						

CTF: Candidate core clock transcription factor. ^1^ “Kafe shift” is the number of h the *K. fedtschenkoi* gene expression profile shifted from its *A. thaliana* ortholog’s gene expression profile. The shift was calculated by subtracting the phases calls of each ortholog. ^2^ “Spear shift” is the Spearman rank correlation coefficient between orthologs after shifting the *K. fedtschenkoi* gene expression profile by the “Kafe shift”.

**Table 2 cells-10-02217-t002:** Clock-controlled stomata-related genes with rescheduled gene expression from different studies.

Clock TF	Relationship ^1^	Study	Target	Stomata-related ^2^	Gene Description
Kaladp0748s0043 (CTF1)	Represses	[21]	Kaladp0059s0048	New	aquaporin pip1-2
Kaladp0748s0043 (CTF1)	Represses	[8]	Kaladp0062s0167	New	receptor-like protein kinase haiku2
Kaladp0007s0017 (CTF2)	Activates	[21]	Kaladp0011s0363	New	trehalose-phosphate synthase
Kaladp0007s0017 (CTF2)	Activates	[8]	Kaladp0092s0084	Known/GO	calcium-dependent protein kinase 26
Kaladp0011s1342 (CTF3)	Represses	[21]	Kaladp0040s0264	New	btb poz domain-containing protein npy2-like
Kaladp0011s1342 (CTF3)	Represses	[21]	Kaladp0008s0539	New	mitogen-activated protein kinase
Kaladp0011s1342 (CTF3)	Represses	[21]	Kaladp0033s0113	Known/GO	phototropin-2
Kaladp0011s1342 (CTF3)	Represses	[8]	Kaladp0001s0016	New	catalase isozyme 1
Kaladp0011s1342 (CTF3)	Represses	[8]	Kaladp0093s0030	New	homeobox-leucine zipper protein anthocyaninless 2 isoform x1
Kaladp0059s0037 (ELF4)	Represses	[8]	Kaladp0062s0076	New	3-ketoacyl- synthase 19-like
Kaladp0047s0123 (LNK1)	Activates	[8]	Kaladp0008s0414	New	cyclic nucleotide-gated ion channel 15
Kaladp0060s0264 (LNK2)	Activates	[8]	Kaladp0024s0371	New	pectin lyase-like superfamily protein isoform1
Kaladp0099s0129 (LNK2)	Activates	[8]	Kaladp0042s0353	Known	abscisic acid receptor pyl8-like
Kaladp0060s0264 (LNK2)	Activates	[8]	Kaladp0095s0634	GO	mitogen-activated protein kinase homolog mmk2
Kaladp0101s0041 (PRR7)	Represses	[8]	Kaladp0092s0115	New	pleiotropic drug resistance protein 1-like
Kaladp0055s0349 (RVE6)	Activates	[8]	Kaladp0043s0103	New	phospholipid-transporting atpase 3
Kaladp0055s0349 (RVE6)	Activates	[8]	Kaladp0090s0003	New	receptor-like protein kinase

CTF: Candidate core clock transcription factor. ^1^ “Relationship” refers to the type of transcriptional regulation. ^2^ “Stomata-related” refers to if the gene is either known as a stomata-related gene via publication in the literature (known), is annotated as a stomata-related (GO) or was identified in [8] as being stomata-related (New).

## Data Availability

All datasets generated for this study are included in the manuscript and the supplementary files.

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
