# Peer review of "Inference of Gene Regulatory Network Uncovers the Linkage between Circadian Clock and Crassulacean Acid Metabolism in Kalanchoë fedtschenkoi"

_cells, 2021, doi:10.3390/cells10092217_

Round 1
Reviewer 1 Report
In this manuscript, Moseley et al., predicted a set of genes that could be components of either the core clock or the networks attached to the core clock in the CAM species Kalanchoë fedtschenkoi. By using the Local Edge Machine (LEM) algorithm, they also inferred stomata-related gene targets for known and candidate core clock genes and constructed a gene regulatory network for core clock and stomata-related genes. This is an interesting manuscript that links circadian clock with crassulacean acid metabolism (CAM) photosynthesis, and provide an insight on how circadian clock controls CAM related genes which may be useful for crop design breeding. Additionally, they also predicted a few candidate core clock transcription factors, which awaits to be further functional validated. I have a few concerns listed as follows.
- Since the type of this manuscript is a research article, so it is somehow unconventional that Figure 1 is presented in the introduction section. All of the figures should be in the section of results obtained from their study. I suggest them move the current figure 1 as a supplementary material.
- Phylogenetic analysis may be needed to analyze the evolutionary relationships between predicted circadian clock related genes in Kalanchoë fedtschenkoi with their corresponding homologies in other species, such as in Arabidopsis.
- In Figure 2, it is unclear that they defined the activation or repression of gene expression, based on which kind of criteria?
- In Figure 3B, what does “CTF” stands for and how did the author identified these seven candidates. These critical results are not addressed in the results section. Figre 3B is first cited in discussion section. All the results should be presented in the order of the Figures in the manuscript.
- The expression profiles of predicted circadian clock related genes in Kalanchoë fedtschenkoi should be at least validated by RT-qPCR under both diurnal and circadian conditions.
- In Figure 7, what do the colors and the size of the circles represent? It would be better to spell them out.
Reviewer 2 Report
This is a straightforward and well-written study in which the authors apply several approaches they developed elsewhere on already published datasets. The study is limited to bioinformatics analysis and generates a list of candidate genes, wihtout any experimental validation/exploration. Nevertheless, it is an informative study opening up many new hypotheses.
I only have a couple of minor comments
Abstract
- The authors use here (and elsewhere) 'a new metric', although the DLxJTK metric has been published by the authors in a preprint elsewhere (Motta et al). I suggest replacing 'new' with DLxJTK. I was under the impression that this paper would introduce a new metric with this formulation.
Intro
- Make clear that with "movement', opening and closure of stomata is meant and introduce this a bit more in relation to CAM.
Main
- Figure 2 and related paragraph. Introduce both better. If I'm correct you used LEM to reveal these relationships. The paragraph and figure should start with explaining this.
- Figure 3. Core clock TF in first sentence: put (CTF1-7) after that and/or explain the abbreviation CTF in the figure.
- Table 1. Add also CTF number to this table.
- MYC2 is well-known to regulata stomatal opening and closure. This could be discussed here.
Minor textual
- CO2 subscript 2
- Italics for species names
Round 2
Reviewer 1 Report
The authors have addressed my former concerns. I have no other issues.